# Integration of Chemometrics and Sensory Metabolomics to Validate Quality Factors of Aged Baijiu (Nianfen Baijiu) with Emphasis on Long-Chain Fatty Acid Ethyl Esters

**DOI:** 10.3390/foods12163087

**Published:** 2023-08-17

**Authors:** Yashuai Wu, Hao Chen, Yue Sun, He Huang, Yiyuan Chen, Jiaxin Hong, Xinxin Liu, Huayang Wei, Wenjing Tian, Dongrui Zhao, Jinyuan Sun, Mingquan Huang, Baoguo Sun

**Affiliations:** 1China Food Flavor and Nutrition Health Innovation Center, Beijing Technology and Business University, Beijing 100048, China; wyss995418706@163.com (Y.W.); 2130022063@st.btbu.edu.cn (H.C.); sunyue_991129@163.com (Y.S.); huanghe3938@163.com (H.H.); chenyiyuan_1112@163.com (Y.C.); m15205199608@163.com (J.H.); liuxinxin2580123@163.com (X.L.); 13563938331@163.com (H.W.); sunjinyuan@btbu.edu.cn (J.S.); huangmq@th.btbu.edu.cn (M.H.); sunbg@btbu.edu.cn (B.S.); 2Key Laboratory of Brewing Molecular Engineering of China Light Industry, Beijing Technology and Business University, Beijing 100048, China; 3Beijing Laboratory of Food Quality and Safety, Beijing Technology and Business University, Beijing 100048, China; 4Department of Nutrition and Health, China Agriculture University, Beijing 100193, China; 5Department of Food and Bioengineering, Beijing Vocational College of Agriculture, Beijing 102442, China; 91505@bvca.edu.cn

**Keywords:** aged Baijiu (nianfen Baijiu), long-chain fatty acid ethyl ester, chemometrics, sensomics, softness

## Abstract

The storage process of Baijiu is an integral part of its production (the quality undergoes substantial changes during the aging process of Baijiu). As the storage time extends, the flavor compounds in Baijiu tend to undergo coordinated transformation, thereby enhancing the quality of Baijiu. Among them, long-chain fatty acid ethyl esters (LCFAEEs) were widely distributed in Baijiu and have been shown to have potential contributions to the quality of Baijiu. However, the current research on LCFAEEs in Baijiu predominantly focuses on the olfactory sensation aspect, while there is a lack of systematic investigation into their influence on taste and evaluation after drinking Baijiu during the aging process. In light of this, the present study investigates the distribution of LCFAEEs in Baijiu over different years. We have combined modern flavor sensory analysis with multivariate chemometrics to comprehensively and objectively explore the influence of LCFAEEs on Baijiu quality. The results demonstrate a significant positive correlation between the concentration of LCFAEEs and the fruity aroma (*p* < 0.05, r = 0.755) as well as the aged aroma (*p* < 0.05, r = 0.833) of Baijiu within a specific range; they can effectively reduce the off-flavors and spicy sensation of Baijiu. Furthermore, additional experiments utilizing a single variable suggest that LCFAEEs were crucial factors influencing the flavor of Baijiu, with Ethyl Palmitate (EP) being the most notable LCFAEE that merits further systematic investigation.

## 1. Introduction

Baijiu, the Chinese national liquor, is considered a cultural treasure. Baijiu, a cherished and distinctive alcoholic beverage in Chinese culture, is a traditional Chinese distilled liquor produced through the process of fermentation and distillation using grains such as sorghum, wheat, or rice. The production process encompasses vital steps, starting from the selection of raw materials to saccharification, followed by distillation, storage, and blending. Each step plays a crucial role in enhancing the intricate flavor, aroma, and overall quality of Baijiu. Over thousands of years, it has become an integral part of the lives of the Chinese people.

One of the prominent characteristics of Baijiu is that its quality improves with extended storage time. As a result, the duration of storage has gained significant attention from the public [1,2]. In China, aged Baijiu (nianfen Baijiu) is defined as raw Baijiu that has undergone a longer storage period. The primary drawback of raw Baijiu lies in its inconsistent aroma and sharp, stimulating mouthfeel. In contrast, aged Baijiu is highly favored by consumers due to its pleasant aroma and harmonious taste. The production process of aged Baijiu begins with the careful selection of high-quality brewing materials [3,4]. Refined brewing techniques are then employed to achieve the ideal structure and aroma for Baijiu. Through extensive storage, the trace components in Baijiu interact with each other, resulting in a harmonious balance between taste and aroma. Concurrently, as the storage time increases, it strengthens the flavor profile and prolongs the aroma persistence of the Baijiu, while reducing the presence of impurities and off-flavors [4,5,6]. Aged Baijiu offers a deep and unique flavor, enabling people to appreciate the nuances and changes that occur over time by sampling different years of Baijiu. It is not only a testament to traditional craftsmanship and culture but also a result of continuous innovation within the Baijiu industry, providing consumers with a wide range of choices and a heightened level of enjoyment [7,8,9].

Currently, research on aged Baijiu primarily focuses on analyzing its aroma compounds and comparing the variations in trace components between different years. It has been extensively researched and proven that the trace components (e.g., esters, alcohols, and aromatic compounds) have a remarkable influence on the aroma of Baijiu. For instance, ester compounds in Baijiu mainly exhibit fruity aromas, and acid compounds present cheese flavor [1,4,6]. Based on this, studies primarily concentrate on the correlation analysis between the variations in trace components during the storage process of aged Baijiu and their influence on its aroma. It is important to note that during the storage process, the quality changes of aged Baijiu were not solely confined to alterations in the aroma dimension. While the taste and evaluation after drinking Baijiu hold equal significance for evaluation purposes [10]. Research findings indicated that the pronounced influence of LCFAEEs on the quality of aged Baijiu was verified by combining the distribution of LCFAEEs and sensory evaluation for different years of Baijiu. However, due to the limited number of experimental samples and the years typically spanning 3–5 years, it was not possible to comprehensively reflect the influence of LCFAEEs on the quality of aged Baijiu [11,12,13].

Currently, the essential impact of specific compounds on the aroma profile of Baijiu has been revealed through modern flavor sensomics. Through GC-MS analysis, a specific Baijiu type was found to contain 126 aroma substances. Component omission experiments revealed ethyl hexanoate and ethyl acetate as the primary aroma compounds, while butyric acid, hexanoic acid, ethyl butyrate, ethyl octanoate, 3-methylbutyl hexanoate, and phenethyl alcohol also played significant roles in shaping the aroma profile [14]. Unfortunately, recombinant Baijiu samples cannot simulate the complex chemical composition and interactions found in real Baijiu, which may lead to discrepancies between experimental results and actual situations [15,16,17]. Accordingly, this study aims to clarify the significant impact of LCFAEEs in Baijiu on the expression of Baijiu quality by employing a multidimensional chemometric approach in conjunction with sensory group analysis.

Based on the study’s main focus, six commonly found LCFAEEs were selected, namely ethyl tetradecanoate (ET), ethyl palmitate (EP), ethyl octadecenoate (EO), ethyl 9-octadecenoate (9-EO), ethyl 9, 12-octadecadienoate (912-EO), and ethyl 9, 12, 15-octadecatrienoate (91215-EO). Specifically, samples of Baijiu aged for 30 years from a specific distillery were collected to define the distribution of LCFAEEs. Based on the quality evaluation system (i.e., visual sense, olfactory sense, taste sense, and evaluation after drinking) for Baijiu, the correlation between LCFAEEs and the changes in Baijiu quality during the aging process was elucidated by multivariate chemometrics and molecular sensory metabolomics. Afterwards, a univariate variable addition model for Baijiu was constructed to screen for key markers in aged Baijiu.

## 2. Materials and Methods

### 2.1. Standard Information

The veracity and provenance of standardized information are presented in Table 1 below.

### 2.2. Baijiu Sample Information

The Baijiu samples used in Table 2 as aged beverages were obtained from a Baijiu distillery that manufactured them from 1993 to 2018. Please note that the parallel samples of Baijiu used in this experiment were sourced from separate storage tanks to assess the ethanol content and for sample injection, ensuring the reliability of the experimental data (*n* ≥ 10). It was worth mentioning that the sources of the Baijiu samples mentioned in this article were undisclosed, and the samples were strictly intended for research purposes and not for any commercial use.

### 2.3. Preprocessing Methods for Baijiu Samples

The Baijiu sample was vigorously agitated and then left undisturbed for 10 min. Afterwards, it was filtered and purified using a 0.22 mm filter membrane sourced from Beijing InnoChem Science and Technology Co., Ltd. (Beijing, China). The resultant filtered sample was subsequently analyzed using the designated analytical instrument. To guarantee precision and maintain the reliability of the acquired results, this entire procedure was replicated three times for each individual sample.

### 2.4. Analytical Parameters for GC-MS Detection

In this study, the analytical instruments utilized were the Agilent Technologies 7890B GC System and the Agilent Technologies 5977A MSD (Agilent Technologies Inc., Santa Clara, CA, USA). The injection of each sample was conducted in split-less mode with a 1 μL volume introduced into the system. To ensure cross-verification of the Retention Indices (RIs) of the samples, a DB-WAX column (sourced from J&W Scientific, Folsom, CA, USA) with dimensions of 60 m × 0.25 mm i.d. and a 0.25 μm film thickness was employed for sample analysis. Helium gas was used as the carrier gas, flowing continuously at a rate of 1.5 mL/min. The injector temperature was maintained at 250 °C. The temperature program in the oven was as follows: the oven temperature was initially set at 40 °C and then increased to 200 °C at a rate of 30 °C/min, remaining constant for 2 min. Subsequently, the temperature was further raised at a rate of 2 °C/min to reach 240 °C and maintained for 10 min.

In the mass spectrometry (MS) analysis (Agilent Technologies Inc., Santa Clara, CA, USA), we employed an electron ionization (EI) mode with an ionization energy of 70 eV. To initiate the activation process, the voltage was adjusted to a precise value of 1.5 V. To compensate for the solvent delay, the initiation of MS acquisition occurred within a time frame of 4–8 min following sample injection. The interface temperature was consistently maintained at 250 °C, while the ion source temperature was precisely set to 230 °C. For the purpose of identifying LCFAEEs, a comprehensive full scan mode was employed, encompassing a mass range from 45 to 350 atomic mass units (AMU). The quantification analysis of LCFAEEs was performed utilizing the selected-ion-scan mode (SIM).

### 2.5. Qualitative Analysis of LCFAEEs

The aforementioned analytical conditions were used to conduct a comprehensive scan analysis on Baijiu samples. Qualitative analysis was performed by matching the retention times of LCFAEEs in the Baijiu samples to those acquired from standards. To further support the identification process, a comparative assessment was performed by matching the standard mass spectra available in the NIST 20 library. The NIST 20 library is a comprehensive collection of mass spectrometry data maintained by the National Institute of Standards and Technology (NIST). It serves as a vital resource for scientists and researchers in the field of analytical chemistry. With over 275,000 spectra and 178,000 compounds, it offers an extensive database for identifying and characterizing chemical compounds with exceptional accuracy, focusing on the degree of matching and characteristic ions. This analytical approach successfully enabled the identification and characterization of LCFAEEs in the Baijiu samples.

### 2.6. Quantitative Analysis of LCFAEEs

To initiate the analysis of the Baijiu sample, it underwent vigorous shaking followed by a 10 min standing period. Subsequently, a 1 μL volume of the prepared sample was injected into the gas chromatography (GC) system in the split-less mode. The injection conditions used were identical to those outlined in Section 2.4 for LCFAEE identification. The detector temperature was maintained at 250 °C.

For the creation of calibration curves, stock solutions containing a mixture of LCFAEE standards were prepared in absolute ethanol. These solutions were then diluted to obtain a range of concentrations using 60° ethanol as the working standard solution. The resulting working standard solutions were subjected to GC-MS analysis using a gas chromatography-mass spectrometer. A linear regression analysis was performed, with the mass concentration of the solution plotted on the abscissa and the corresponding peak area on the ordinate. Based on the calibration curves, the concentrations of the LCFAEEs in the Baijiu samples were calculated. To determine the limits of detection (LOD), the lowest concentrations of the analyte standard solutions were analyzed with a signal-to-noise ratio of 3. Similarly, the limit of quantitation (LOQ) was determined using a signal-to-noise ratio of 10 [6]. It is important to note that all analyses were conducted in triplicate to ensure reproducibility.

To assess the precision of the analyses, precision tests were conducted for the six LCFAEEs at specific concentrations. Inter-day and intra-day precision analyses were performed to ensure the accuracy and reliability of the results.

### 2.7. Evaluation of Sensomics for Modern Flavor Analysis

The identification of the distinctive style and quality attributes of Baijiu in this study aligns with the principles of international sensory evaluation in the field of food science [18]. This multifaceted approach encompasses a combination of sensory, physical, and chemical experiments. By conducting this comprehensive analysis, valuable insights were gained into the intricate relationship between these compounds and the resulting sensory attributes of Baijiu.

#### 2.7.1. Baijiu Evaluation Environment and Conditions

The evaluation room was meticulously maintained to provide an environment free from vibrations and noise, creating a serene atmosphere conducive to sensory analysis. The room was consistently cleaned and meticulously organized, ensuring the absence of any foreign odors. Moreover, the temperature in the room was carefully regulated within the range of 15–20 °C, offering optimal conditions for the evaluation process.

The Baijiu samples were carefully stored at a controlled temperature of 20 ± 1 °C. Each sample was assigned a unique identifier and organized accordingly to ensure efficient organization and subsequent analysis. The Baijiu evaluations took place during specific time periods, primarily between nine and eleven o’clock in the morning and fifteen and seventeen o’clock in the afternoon. These timeframes were selected with the utmost care, considering optimal sensory perception and evaluator attentiveness. To ensure the relevance and reliability of the sensory experiment results, ten sensory evaluators were selected from the esteemed Key Laboratory of Brewing Molecular Engineering of China Light Industry. These evaluators, aged between twenty-three and twenty-seven, possessed a strong foundation in the subject matter. In addition, another group of ten evaluators from Beijing Technology and Business University, also aged between twenty-three and twenty-seven, were included. This second group did not have relevant professional backgrounds.

#### 2.7.2. Descriptive Sensory Analysis

In this study, a comprehensive sensory evaluation was conducted, encompassing four distinct dimensions: visual sensory, olfactory sensory, taste sensory, and evaluation after drinking. The scoring criteria for each dimension were as follows (Table 3) [19].

##### Visual Sensory

The visual sensory attributes were evaluated using an eleven-point scale, ranging from 0 to 10. This scale allowed for the assessment of visual clarity, ranging from extreme haze (0) to clear transparency (10).

##### Olfactory Sensory

The olfactory sensations were evaluated on a six-point scale, with scores ranging from 0 to 5. This scale facilitated the assessment of aroma intensity, from the absence of aroma (0) to a very strong aroma (5).

##### Taste Sensory

For the taste sensory evaluation, an eleven-point scale was used, spanning from 0 to 10. This scale allowed for the assessment of various aspects, including the presence or absence of aroma (0 representing no aroma) and the intensity of pungent stimuli (10 denoting extremely pungent stimuli with a strong aroma). Additionally, the evaluation encompassed the perception of a soft liquor taste.

##### Evaluation after Drinking

This dimension was gauged on a scale ranging from 0 to 10, assessing the retention of the Baijiu in the mouth. A score of 0 indicated no noticeable retention in the mouth, while a score of 10 signified noticeable retention.

### 2.8. Statistical Examination and Analysis

In our research methodology, we employed various software tools for different analytical purposes. Radar diagrams and fingerprints were generated using Origin 2018 software, provided by Origin Lab Corporation (Northampton, MA, USA). Heatmaps were created using Tbtools, developed by Beijing Liu Zhi Information Technology Co., Ltd. (Beijing, China). Orthogonal partial least squares discriminate analysis (OPLS-DA) and predictor variable importance in projection (VIP) were conducted using SIMCA 14.1 software, produced by Sartorius Stedim (Göttigen, Germany). Single-factor and correlation analyses were performed utilizing SPSS 24.0 software, developed by International Business Machines Corporation (Chicago, IL, USA) [20,21].

## 3. Results and Discussion

### 3.1. Description of Parameters

Table 4 illustrates the selected ion monitor (SIM) parameters utilized for both qualitative and quantitative analysis of a specific set of six LCFAEEs. To ensure the reliability of the quantification method, a linear regression analysis was performed. In this analysis, the mass concentration of the solution was the independent variable (abscissa), while the corresponding peak area served as the dependent variable (ordinate). The outcome of the linear regression analysis led to the development of a standard curve, as detailed in Table 4. This table provides essential information pertaining to the linearity range, regression equations, limits of detection (LOD), and limits of quantification (LOQ) for each LCFAEE. In addition, we closely monitored and documented the intra-day and inter-day precisions for the six LCFAEEs. The intra-day precisions were found to be less than 3%, and the inter-day precisions remained under 9%. Moreover, the recovery percentages, ranging from 95 to 108%, were also recorded in Table 4.

Overall, these findings collectively demonstrate the exceptional accuracy and precision of the proposed analytical method in detecting the targeted LCFAEEs. The method effectively meets the analytical requirements of the investigation.

### 3.2. Distribution of Six LCFAEEs in Baijiu for Different Years

In order to investigate the underlying mechanism behind the impact of the production year on the distribution patterns of LCFAEEs in Baijiu, a comprehensive analysis was conducted on a series of Baijiu sample groups (B-1993, B-2003, B-2008, B-2013, B-2014, B-2015, B-2017, and B-2018). The concentrations of LCFAEEs were measured, and it was found that LCFAEEs were widely distributed among all the Baijiu sample groups; all six types of LCFAEEs were detected. The concentration of 91215-EO was not specified due to its low concentration below the quantification limit. Additionally, significant differences (*p* < 0.05) were observed in the distribution of the remaining five LCFAEEs in the Baijiu samples.

In Baijiu, the concentration of EP was the highest among the five quantified LCFAEEs, and its concentration changed significantly with the year. The concentration of EP in different sample groups ranges from 1.97 to 7.82 mg/L, and the highest concentration was in the B-2003 group (7.82 mg/L), which was significantly different from other sample groups (*p* < 0.05). The concentration of EP in the B-1993 group was relatively high (4.28 mg/L) and significantly different from other sample groups (*p* < 0.05). All in all, while the differences in EP concentration between different sample groups were relatively low, the concentration of EP among the groups was statistically significant (Figure 1a).

The concentrations of 9-EO and 912-EO in Baijiu were only surpassed by EP, and the concentration distribution of these two compounds was similar among different years of Baijiu. Ranging from 1.15 to 2.44 mg/L, the concentration of 9-EO varies across the different sample groups, highlighting the B-2003 group as the one with the highest concentration (2.44 mg/L). This discrepancy was notably significant when compared to the other sample groups (*p* < 0.05). The concentration of 9-EO was relatively low in the B-2013 and B-2015 sample groups (1.15 mg/L), which was significantly different from other sample groups (*p* < 0.05).

Among the various sample groups, the concentration of 912-EO exhibits a range of 1.13 to 3.48 mg/L. Notably, the B-2003 group records the highest concentration (3.48 mg/L), establishing a statistically significant difference from the other sample groups (*p* < 0.05). The concentration of 912-EO was the lowest in the B-2013 sample group (1.13 mg/L), which was significantly different from other sample groups (*p* < 0.05). The concentration of 912-EO in other sample groups ranges from 1.24 to 1.54 mg/L (Figure 1b,c). The low-concentration LCFAEEs in Baijiu were ET and EO.

The concentration of ET varied between 0.33 and 0.52 mg/L across different sample groups. The B-2003 group had the highest concentration (0.52 mg/L), which showed significant differences compared to other sample groups (*p* < 0.05). The B-1993 group exhibited a relatively higher concentration of ET (0.41 mg/L) compared to other sample groups (B-2008, B-2013, B-2014, B-2015, B-2017, and B-2018), demonstrating significant differences (*p* < 0.05). The concentration of ET among the B-2008, B-2013, B-2014, B-2015, B-2017, and B-2018 sample groups did not show significant differences (*p* > 0.05), ranging from 0.33 to 0.37 mg/L (Figure 1d).

EO only reached the detection limit in four Baijiu sample groups with longer storage times, indicating a notable point. Across different sample groups, the concentration range of EO spanned from 0.00 to 1.01 mg/L. Some sample groups (B-2013, B-2015, B-2017, and B-2018) did not reach the quantitation limit, while the remaining four sample groups have very similar concentrations of EO, ranging from 0.96 to 1.01 mg/L. Based on this, it can be inferred that the concentration of EO in Baijiu was relatively low, and its concentration was greatly influenced by the storage time and conditions (Figure 1e).

Based on the above, it can be found that there was a significant difference in the concentration of LCFAEEs in Baijiu between different years. Additionally, the distribution of concentrations for different LCFAEEs in Baijiu from distinct groups corresponds to the cumulative concentration distribution of the five LCFAEEs in Baijiu (Figure 1f). Based on this, the differential analysis of Baijiu samples from different years will take the total concentration of the five LCFAEEs in Baijiu as the dependent variable.

The concentration of LCFAEEs in Baijiu showed an increasing trend within 20 years with the increase in storage time. However, after 20 years, the concentration of LCFAEEs in Baijiu decreased with the increase in storage time. The concentration fluctuations observed in LCFAEEs during the storage of Baijiu can be attributed to the intricate interplay between formation and degradation processes. Initially, esterification reactions may stimulate an increase in the concentration levels of these compounds [22,23,24,25,26,27]. However, as aging goes on, oxidation and hydrolysis gradually diminish their abundance (Figure 2a).

An OPLS-DA analysis was conducted to further explore the significance of the five LCFAEEs in Baijiu from different years. As a result, it was found that the concentration of LCFAEEs was one of the important indicators for Baijiu groups from different years (Figure 2c). Combining the VIP score discrimination, it was revealed that EO, 9-EO, and 912-EO were important markers for Baijiu from different years, which may have a significant influence on the quality of Baijiu (Figure 2b).

Based on a comprehensive understanding of the distribution of LCFAEEs in Baijiu across different years (Appendix A), a thorough analysis was conducted to evaluate the influence of these compounds on sensory quality using modern evaluation methods. This analysis encompassed four key dimensions: visual perception, olfactory perception, taste perception, and post-consumption evaluation.

### 3.3. Analysis of Baijiu Samples Using Modern Flavor Sensomics

The concentration profiles of LCFAEEs in different Baijiu sample groups formed the foundation for an extensive investigation into their influence on Baijiu quality. To confirm the flavor profile of Baijiu samples, subject groups were utilized in conjunction with modern flavor sensomics. In order to showcase a comprehensive representation of the Baijiu flavor, four dimensions, namely visual, olfactory, taste, and post-drinking evaluation, were thoughtfully selected [28,29,30,31,32,33].

By employing a multivariate chemical analysis method within this framework, the connection between LCFAEEs and the sensory dimensions of Baijiu was further explored. This approach led to a refined and detailed comprehension of the associations between LCFAEEs and the sensory aspects of Baijiu (Appendix A).

#### 3.3.1. Visual Sense

There were significant differences (*p* < 0.05) in the visual evaluation scores among the different Baijiu sample groups. The relatively lower scores were due to the presence of slight turbidity in the B-1998 and B-2003 groups, and the B-2008 group had lower scores compared to other groups (B-2013, B-2014, B-2015, B-2017, and B-2018). Based on this, it can be concluded that as the storage time of Baijiu increased, it tended to become turbid with a slight color change (Figure 3). This change could be a result of the natural degradation and oxidation processes that occur over time. The interaction of various compounds in Baijiu, such as phenolic compounds and aging-related substances, may lead to the formation of sediments or precipitates, giving rise to turbidity. Additionally, the oxidation of certain components may contribute to the coloring of the Baijiu body. Furthermore, prolonged storage may also facilitate the hydrolysis of esters and the breakdown of complex molecules, potentially releasing additional compounds that impact the visual appearance of Baijiu [34,35,36].

#### 3.3.2. Olfactory Sense

The sensory scores of the five dimensions exhibited significant variations among different samples of Baijiu (*p* < 0.05), aligning with previous studies (Figure 4) [37,38,39]. Moreover, with an increase in the storage time of Baijiu, the aroma of floral, fruity, and aged notes exhibited a heightened intensity (Figure 4a,b,e). By combining the concentration of LCFAEEs with the aroma scores according to Baijiu among different sample groups, it can be noted that there was a significant positive correlation between the concentration of LCFAEEs and fruity aroma (*p* < 0.05, r = 0.755), as well as aged aroma scores (*p* < 0.05, r = 0.833), whereas no significant correlation was found with other aroma dimensions (Figure 4). The significant differences in sensory expression among different years of Baijiu can be attributed to various factors, mainly aging conditions. The consistency of these findings with numerous studies suggests that these differences in sensory scores were not isolated incidents but rather a common phenomenon in the Baijiu industry [40,41,42,43].

Through the analysis of the correlation between different aroma dimensions, it can be observed that there was a negative correlation between the sweet-smelling of the Baijiu and the aroma scores in the floral, fruity, and grain dimensions (Figure 4f). Combined with relevant studies, it was inferred that there was a potential interaction between aroma compounds with floral, fruity, and grain aromas (e.g., esters and aromatic compounds) and aroma compounds with sweet-smelling (e.g., alcohols) [44,45,46].

Combined with the above Baijiu aroma expression, it can be found that LCFAEEs were one of the factors affecting the aroma of Baijiu. Nevertheless, the flavor characteristic of Baijiu was determined by the trace components present in different types of Baijiu [47,48,49,50]. Numerous studies have shown that the aroma of Baijiu is mainly determined by low-molecular-weight compounds, including esters, alcohols, and acids. While the contribution of LCFAEEs to the aroma of Baijiu was not significant due to their high odor thresholds, as confirmed by this study. To sum up, it was the interplay between these trace components that ultimately shaped the distinct aroma profile of the Baijiu [51,52,53].

#### 3.3.3. Taste Sense

This sensory dimension was designed based on the typical drinking sequence of individuals (i.e., entry, oral cavity, and throat) to accurately replicate the drinking experience and collect the most genuine data. For the dimension of initial gustatory impression, the B-1993 sample group was significantly higher than the other sample groups (*p* < 0.01), indicating that with the increase in storage time, the body of the B-1993 Baijiu samples was noticeably better than the other sample groups, resulting in a smoother mouthfeel upon entry (Figure 5a). Additionally, the lower scores among the other sample groups indirectly confirm the age-old proverb “Baijiu gets better with age” [9,54]. As for the dimension of olfactory sensation in the oral cavity, the B-1993 sample group showed significant superiority over the other sample groups (*p* < 0.05) (Figure 5b). For the dimension of mid-laryngeal gustatory perception, there were significant differences in ratings between different sample groups, with the B-2003 sample group demonstrating a noticeably better sensation upon entry to the throat compared to the other sample groups (*p* < 0.05). Combined with the result for the distribution of LCFAEEs, it was likely due to the significant contribution of LCFAEEs to the “softness” (i.e., reduced spiciness upon entry) of the Baijiu (Figure 5c).

#### 3.3.4. Evaluation after Drinking

By comparing the evaluation after drinking scores of different aged Baijiu samples, it was found that the perception of stimulation in the B-1993 sample group was significantly lower than in the other sample groups (*p* < 0.01). By examining the concentration of LCFAEEs in different sample groups, it was obvious that, the concentration of LCFAEEs was positively correlated with the perception of irritation (r = 0.429). It can be deduced that LCFAEEs have a positive effect on the “softness” of the Baijiu. Interestingly, the B-1993 sample group has markedly lower ratings in terms of off-flavor dimensions compared to the other sample groups (*p* < 0.05). The occurrence of this phenomenon may be attributed to prolonged storage time, which can lead to the emergence of off-flavors (i.e., strong, pungent, and unwelcome aromas) during the aftertaste of Baijiu (Figure 6) [4,25,55,56]. In addition, there was relatively little difference in the off-flavor dimension ratings between the B-2003 sample group and the Baijiu samples produced in the past decade. Based on the above, it is evident that while long-term storage can enhance the “softness” of Baijiu to a significant extent, it may also result in an augmentation of “unpleasant flavors” upon consumption. Therefore, aged Baijiu was not necessarily better with longer aging and had an optimal consumption period.

Taking into account the aforementioned sensory experiment, we can preliminarily discuss the quality of different years of Baijiu. However, it is incorrect to assume that the quality of Baijiu improves with an increase in storage time. Prolonged storage time of Baijiu can lead to issues such as turbidity, discoloration, and off-flavors in the Baijiu. Hence, as a beverage, Baijiu does indeed have an ideal period for consumption. Based on the sensory experiment data, we can observe that the quality of Baijiu stored for 10–20 years, overall, is relatively good. However, Baijiu stored for over 20 years may encounter some quality issues. It is important to note that these conclusions are solely drawn from the samples used in this experiment. The sample size does not encompass the entirety of Baijiu products manufactured by distilleries in those corresponding years. Furthermore, it is crucial to acknowledge that this study did not explore whether the quality of Baijiu improves with storage periods surpassing 40, 50, or even longer years. In order to draw more precise conclusions, a substantial number of samples and a wider variety of Baijiu types are needed to provide support.

Based on the preceding analysis, LCFAEEs possess the remarkable ability to enhance the softness of Baijiu. However, whether LCFAEEs have an impact on the quality of Baijiu still requires specific verification, and the individual influence of distinct LCFAEEs on Baijiu quality remains ambiguous. As such, the subsequent phase of this study endeavors to execute sensory evaluations aimed at elucidating whether specific LCFAEEs independently yield significant effects on the sensory attributes of Baijiu.

### 3.4. Baijiu Sample Addition Experiment

Building upon the aforementioned discoveries, it can be clearly stated that LCFAEEs possess significant effects on the flavor quality of Baijiu. To further unravel the intricacies of this interaction, the present study aimed to conduct Baijiu addition experiments, specifically targeting the impact of individual LCFAEEs on Baijiu quality. To ensure optimal experimental feasibility and reliable results, the investigation focused exclusively on assessing the effects of LCFAEEs on the sensory quality of Baijiu, employing a controlled variable methodology. After considering the distribution of LCFAEEs in Baijiu from different years, a sample group of Baijiu with a relatively moderate concentration of LCFAEEs was selected for the single-variable addition experiment. The research group selected the Baijiu from the B-2013 sample group as the basic Baijiu sample. LCFAEEs were added singly to the Baijiu of the B-2013 sample group, so that the total concentration of LCFAEEs in the recombined sample group reached the total concentration of LCFAEEs in the B-1993 sample group. The above sample groups were coded as B-2013-ET, B-2013-EP, B-2013-EO, B-2013-9-EO, and B-2013-912-EO. Five different LCFAEEs were added to the Baijiu of the B-2013, B-2015, and B-2018 sample groups, respectively, so that the concentrations of the five LCFAEEs in the recombined sample groups were exactly the same as those in the B-1993 sample group. The above sample groups were coded as B-2013-1993, B-2015-1993, and B-2018-1993.

The meticulous examination and statistical assessments revealed that there was no statistically significant (*p* > 0.05) difference between the recombinant Baijiu samples (B-2013-ET, B-2013-EP, B-2013-EO, B-2013-9-EO, B-2013-912-EO, and B-2013-1993) and their corresponding basic Baijiu (B-2013) counterparts in terms of visual perception. These results indicate that, from a visual standpoint, the recombinant Baijiu samples closely resemble the original Baijiu samples.

After a single addition of LCFAEEs to the Baijiu sample, analysis of the aroma profile of the Baijiu by OPLS-DA revealed significant changes in the Baijiu aroma (Figure 7a). Based on the VIP discrimination, it can be inferred that the addition of LCFAEEs had the greatest impact on the floral and aged aroma of the Baijiu (*p* < 0.05) (Figure 7b). It was also found that the five different LCFAEEs had a similar influence on the aroma of the Baijiu. Through comparison of the olfactory sensory evaluation, an interesting finding was obtained: the scores of the B-1993 and B-2013-1993 sample groups were more similar (Figure 7c). Therefore, LCFAEEs made an important contribution to the aroma quality of Baijiu, and the addition of LCFAEEs significantly improved the aged and grain aroma of Baijiu (*p* < 0.05). Combining the summary in Section 3.3.2, it was worth considering LCFAEEs as one of the markers for the aged aroma of Baijiu.

After the single addition of LCFAEEs to the Baijiu samples, cluster analysis was conducted on the taste and evaluation after drinking scores of the recombinant Baijiu samples and the basic Baijiu groups. It was found that the five types of LCFAEEs each had an impact on the taste and evaluation after drinking the Baijiu (Figure 8a). Through the utilization of OPLS-DA, it has been discovered that the incorporation of LCFAEEs significantly influences the taste and evaluation after drinking Baijiu (Figure 8b). The dimensions of mid-laryngeal gustatory perception and off-flavors have greater responsibility for Baijiu (Figure 8c). A comparison between the Baijiu samples before and after addition revealed (Figure 8d–f) that the post-addition Baijiu sample had significantly fewer off-flavors than the original Baijiu sample, a decreasing trend in perceived stimulation, and a significant decrease in spiciness upon consumption, but it may reduce the intensity of aroma release in the oral cavity. By comparing the data from the individual addition of LCFAEEs (Figure 8g–k), it was found that EP and 912-EO had a more remarkable effect on the aforementioned characteristics.

The addition of LCFAEEs to Baijiu leads to a significant reduction in off-flavors compared to the basic Baijiu sample. This can be attributed to the unique properties of fatty acid esters, which have the ability to mask or neutralize undesirable flavors present in beverages [57,58]. One noteworthy effect of adding LCFAEEs to Baijiu is that it can notably reduce the spicy sensation experienced upon tasting. This can be explained by the interaction between the esters and the compounds responsible for the spiciness in Baijiu. The esters may act as sensory modulators by inhibiting the activation of certain receptors that perceive spiciness, resulting in a milder and less intense spicy sensation. Although the addition of LCFAEEs offers benefits like flavor improvement and reduced spiciness, it is important to acknowledge that it also results in a decrease in the intensity of aroma release in the oral cavity [11,25,27,54,59,60,61]. This can be attributed to the physical properties of the esters, which may impede the diffusion of aroma compounds and their interaction with receptors in the mouth. As a result, the overall perception of the aroma may be less pronounced.

Based on the above, EP and 912-EO were key factors influencing the flavor characteristics of Baijiu. It was worth noting that the potential synergistic or antagonistic effects between EP and 912-EO and other trace components in Baijiu deserved further investigation.

All in all, LCFAEEs have the potential to enhance the flavor quality of various beverages beyond the Baijiu industry. By utilizing LCFAEEs in a judicious manner, it becomes feasible to effectively adjust the flavor balance and craft beverages that are remarkably soft and enjoyable.

## 4. Conclusions

This study investigated the distribution of LCFAEEs in Baijiu samples spanning 30 years. The distribution of LCFAEEs in Baijiu from different years was clarified. The impact of LCFAEEs on the quality of Baijiu was explored using multivariate chemometrics and modern sensory technology, specifically assessing the visual, olfactory, taste, and after-drinking evaluation dimensions. As a result, it was revealed that there were significant positive correlations between LCFAEE concentration and fruity aroma (*p* < 0.05, r = 0.755), as well as sweet-smelling (*p* < 0.05, r = 0.833). Additionally, higher concentrations of LCFAEEs led to reduced off-flavors, less stimulation, and a significant decrease in spiciness. Combining the results of individual addition experiments and the distribution of LCFAEEs in Baijiu, it can be concluded that the addition of ethyl palmitate (EP), ethyl octadecenoate (EO), and ethyl 9-octadecenoate (9-EO) had a more significant impact on the aroma of Baijiu. Moreover, the addition of ethyl palmitate (EP) and ethyl 9,12-octadecadienoate (912-EO) significantly reduces off-flavors and irritation sensations. Therefore, LCFAEEs play a key role in influencing the flavor of Baijiu, with ethyl palmitate (EP) being the most noteworthy.

## Figures and Tables

**Figure 1 foods-12-03087-f001:**
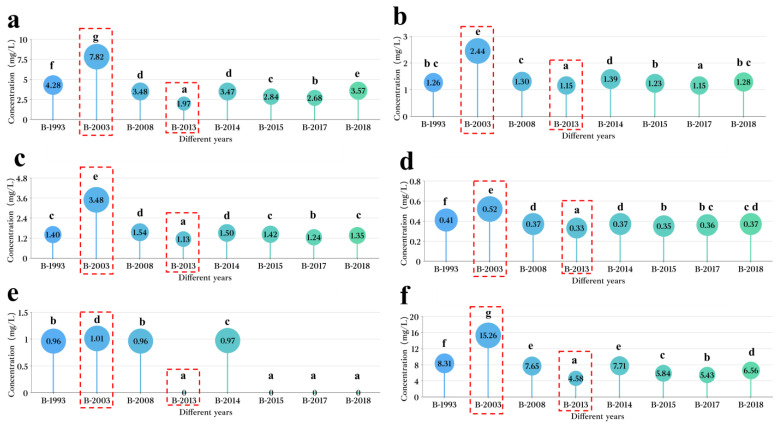
Analysis of the distribution of LCFAEEs in Baijiu from different years. (**a**) Distribution of EP in Baijiu from different years. (**b**) Distribution of 9-EO in Baijiu from different years. (**c**) Distribution of 912-EO in Baijiu from different years. (**d**) Distribution of ET in Baijiu from different years. (**e**) Distribution of EO in Baijiu from different years. (**f**) Distribution of five types of LCFAEEs in Baijiu from different years. The letter (i.e., a) in the figure indicates significance.

**Figure 2 foods-12-03087-f002:**
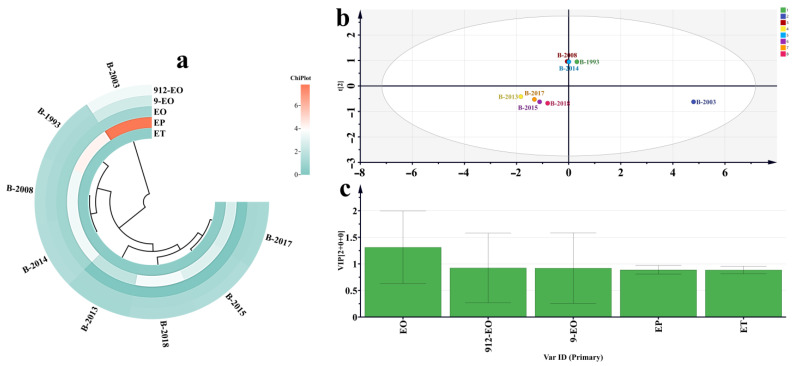
Analysis chart of five LCFAEEs from different years of Baijiu. (**a**) A heatmap depicting the concentrations of five specific LCFAEEs in Baijiu samples from different years. (**b**) Orthogonal partial least square discriminant analysis (OPLS-DA) performed on the concentrations of five LCFAEEs in Baijiu samples from various years. (**c**) Variable importance in projection (VIP) score analysis was used to evaluate the contribution of the concentrations of five LCFAEEs in differentiating Baijiu samples from various years.

**Figure 3 foods-12-03087-f003:**
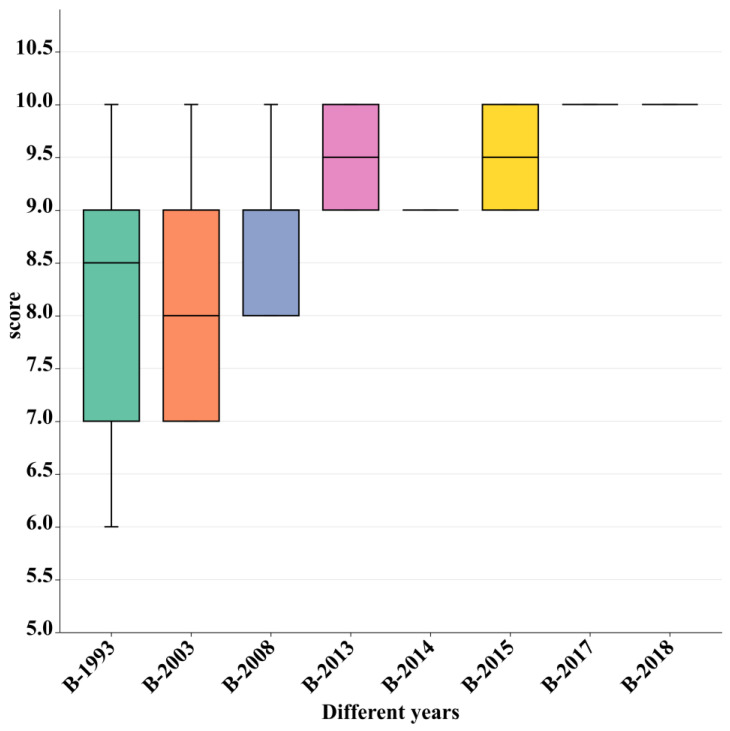
Box diagram of the visual sensory score distribution of different Baijiu sample groups.

**Figure 4 foods-12-03087-f004:**
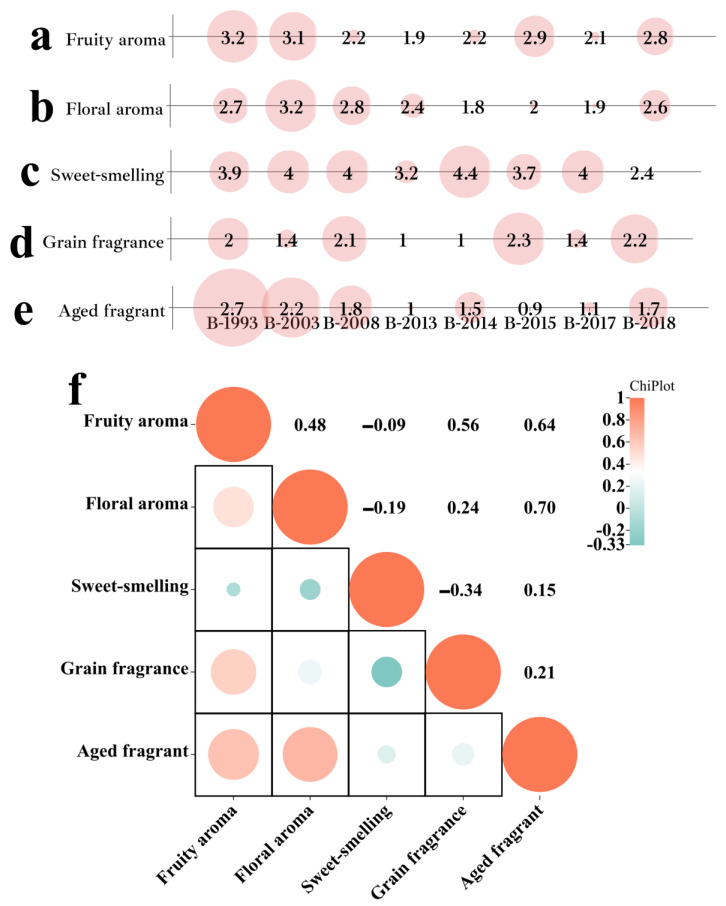
Olfactory sense evaluation of different Baijiu groups. (**a**) Evaluation scores of “fruity aromas” in different Baijiu sample groups. (**b**) Evaluation scores of “floral aromas” in different Baijiu sample groups. (**c**) Evaluation scores of “sweet-smelling” in different Baijiu sample groups. (**d**) Evaluation scores of “grain fragrance” in different Baijiu sample groups. (**e**) Evaluation scores of “aged fragrant” in different Baijiu sample groups. (**f**) Interaction of different aroma dimensions in Baijiu.

**Figure 5 foods-12-03087-f005:**
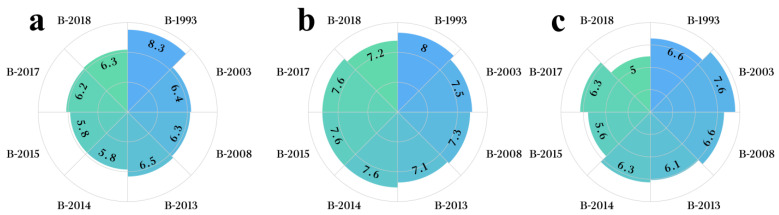
An evaluation of taste and sensory properties among distinct Baijiu groups. (**a**) Initial gustatory impression evaluation of Baijiu in different production years. (**b**) Olfactory sensation in the oral cavity: evaluation of Baijiu in different production years. (**c**) Mid-laryngeal gustatory perception evaluation of Baijiu in different production years.

**Figure 6 foods-12-03087-f006:**
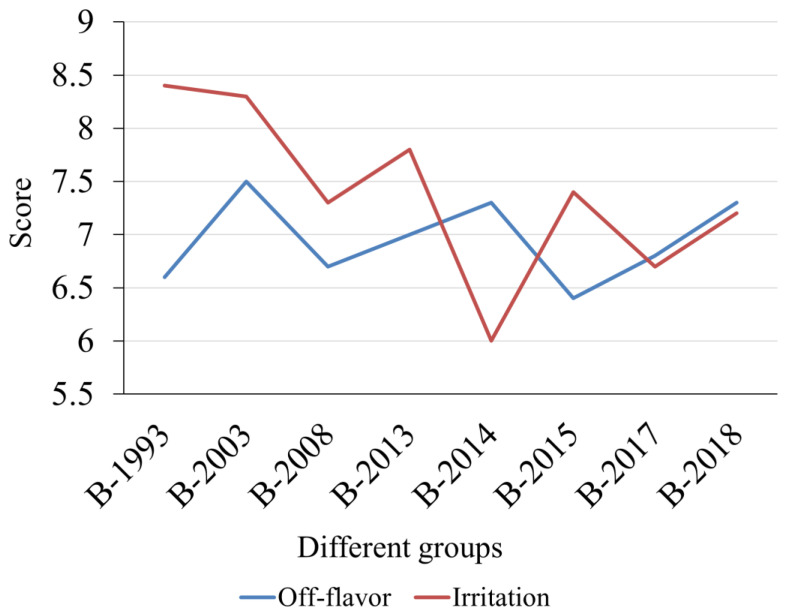
Evaluation after drinking Baijiu samples in different years.

**Figure 7 foods-12-03087-f007:**
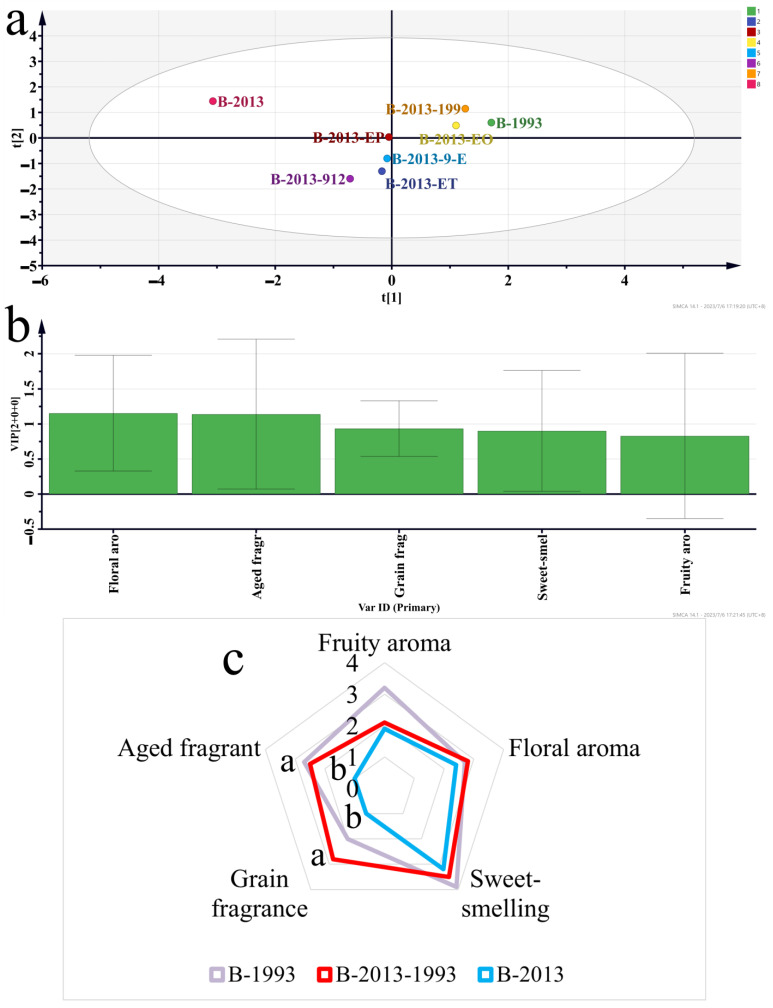
Comparison of olfactory sensory scores between the recombinant sample group and the original samples group. (**a**) OPLS-DA in the sample group. (**b**) VIP in the sample group. (**c**) Comparison of olfactory sensory scores between sample groups B-2013, B-1993, and B-2013-1993. The letter (i.e., a) in the figure indicates significance.

**Figure 8 foods-12-03087-f008:**
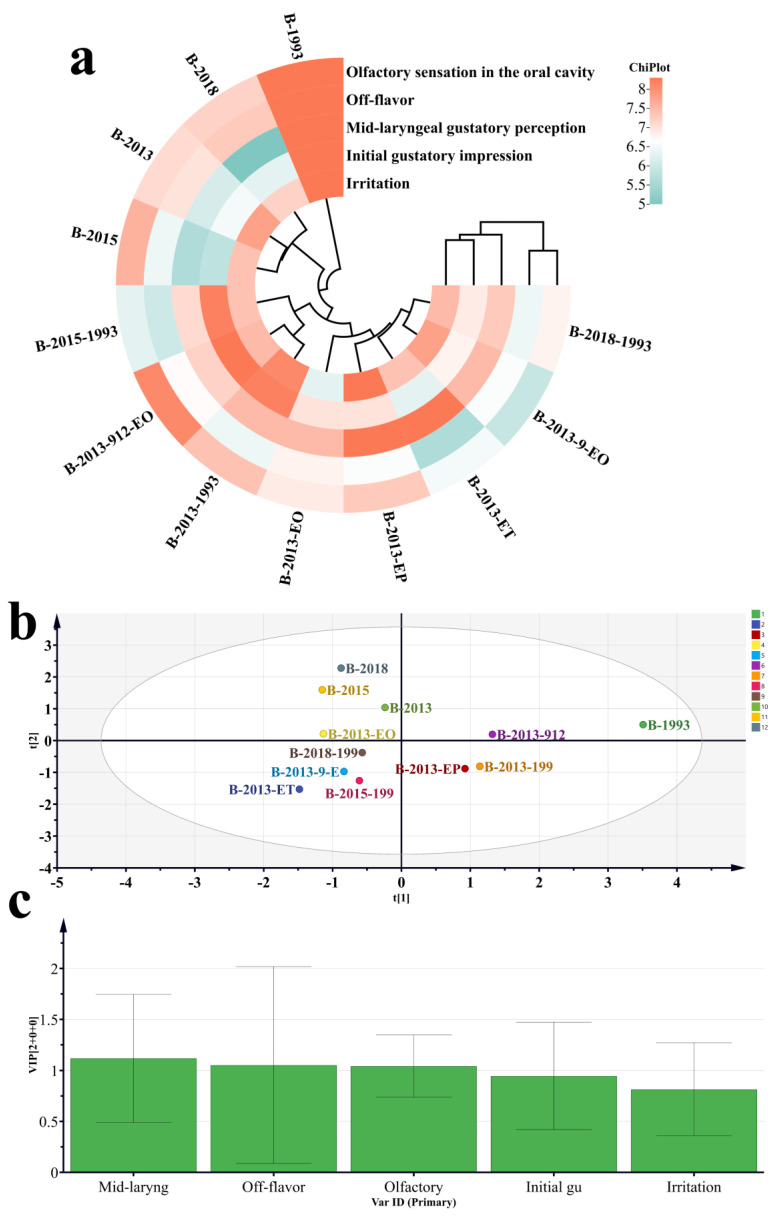
Comparison of taste and evaluation after drinking scores between the recombinant sample group and the original sample group. (**a**) Heatmap of different sample groups. (**b**) OPLS-DA in different sample groups. (**c**) Analysis of VIP in different sample groups. (**d**) Comparison of taste and evaluation after drinking scores between sample groups B-2013 and B-2013-1993. (**e**) Comparison of taste and evaluation after drinking scores between sample groups B-2015 and B-2015-1993. (**f**) Comparison of taste and evaluation after drinking scores between sample groups B-2018 and B-2018-1993. (**g**) Comparison of taste and evaluation after drinking scores between sample groups B-2018 and B-2013-ET. (**h**) Comparison of taste and evaluation after drinking scores between sample groups B-2018 and B-2013-EP. (**i**) Comparison of taste and evaluation after drinking scores between sample groups B-2018 and B-2013-EO. (**j**) Comparison of taste and evaluation after drinking scores between sample groups B-2018 and B-2013-9-EO. (**k**) Comparison of taste and evaluation after drinking scores between sample groups B-2018 and B-2013-912-EO.

**Table 1 foods-12-03087-t001:** Experimental standards, fineness, and sources.

Name	Abbreviation of Chart Name	CAS	Fineness	Sources
Ethyl tetradecanoate	ET	124-06-1	GC ≥ 98.0%	Beijing InnoChem Science and Technology Co., Ltd.
Ethyl palmitate	EP	628-97-7	GC ≥ 98.0%	
Ethyl octadecanoate	EO	111-61-5	GC ≥ 99.0%	
Ethyl 9-octadecenoate	9-EO	111-62-6	GC ≥ 98.5%	
Ethyl 9,12-octadecadienoate	912-EO	544-35-4	GC ≥ 97.0%	
Ethyl 9,12,15-octadecatrienoate	91215-EO	1191-41-9	GC ≥ 98.0%	

**Table 2 foods-12-03087-t002:** Information on experimental Baijiu samples.

Numbering of Baijiu Sample Groups	Descriptive Information about Baijiu	Alcohol/°
B-1993	A certain Baijiu produced in 1993	56
B-2003	A certain Baijiu produced in 2003	58
B-2008	A certain Baijiu produced in 2008	54
B-2013	A certain Baijiu produced in 2013	55
B-2014	A certain Baijiu produced in 2014	59
B-2015	A certain Baijiu produced in 2015	59
B-2017	A certain Baijiu produced in 2017	58
B-2018	A certain Baijiu produced in 2018	60

**Table 3 foods-12-03087-t003:** The sensory evaluation report for Baijiu samples.

One-Level Indicators	Two-Level Indicators	Specific Description	Rating Score (Points)
Visual sense	Clear and transparent with an absence of impurities	Optically pure with no suspension or precipitation.	0–10
Olfactory sense	Fruity aroma	The aroma profile of Baijiu showcases a diverse range of fruity notes, encompassing fragrances reminiscent of various fruits such as banana, apple, pineapple, and more. These fruity aromas yield a pleasant olfactory experience, evoking a sense of comfort and enjoyment.	0–5
	Floral aroma	The sensory attribute referred to as “sweet flower fragrance” encompasses a delightful and appealing floral aroma. This aroma specifically includes the scents of flowers such as roses, chrysanthemums, and others.	0–5
	Sweet-smelling	The mellow aroma of Baijiu is characterized by its smooth, nuanced, and lingering fragrance that evokes a harmonious blend of distinct and complex notes.	0–5
	Grain fragrance	Pleasing aroma of cooked cereal grains.	0–5
	Aged fragrant	Distinct from Baijiu, this aroma can be characterized as an evident oak barrel aroma.	0–5
Taste sense	Initial gustatory impression	The intricate trace constituents in Baijiu may elicit transient, mild irritation.	0–10
	Olfactory sensation in the oral cavity	The profound Baijiu aroma resembles an exhilarating volcanic eruption enveloping the oral cavity.	0–10
	Mid-laryngeal gustatory perception	Absence of throat irritation and Baijiu’s subtle softness.	0–10
Evaluation after drinking	Off-flavor	The off-flavor of Baijiu is characterized by unpleasant and atypical sensory attributes that deviate from the desired qualities, often exhibiting notes of mustiness, earthiness, or pungency.	0–10
	Irritation	The characteristics of Baijiu’s irritation can manifest as a transient, mild sensation that momentarily stimulates the throat without inducing significant discomfort.	0–10

**Table 4 foods-12-03087-t004:** Key parameters of LCFAEEs.

Compounds	Qualitative Ion	Quantitative Ion	LOD/(μg/L)	LOQ/(μg/L)	Intra-Day Precision/(%)	Inter-Day Precision/(%)	Recovery Rate/(%)	Regression Equation	R^2^
ET	88, 101, 157, 211, 256	88, 256	0.24	0.76	1.1	2.57	95–103	y = 307,753x − 80,514	0.9989
EP	88, 101, 157, 239, 284	88, 284	0.32	1.91	1.15	1.73	105–108	y = 308,141x − 217,577	0.9976
EO	88, 101, 157, 269, 312	88, 312	0.11	1.13	2.11	3.79	95–101	y = 1,000,000x − 929,800	0.9896
9-EO	55, 180, 222, 264, 310	55, 310	0.18	1.18	1.13	1.15	96–106	y = 968,421x − 943,401	0.9958
912-EO	67, 81, 109, 263, 308	67, 308	0.24	3.32	2.14	8.62	98–107	y = 1,000,000x − 1,000,000	0.9950
91215-EO	79, 95, 108, 121, 306	79, 306	0.44	1.15	2.35	0.85	102–103	y = 1,000,000x − 1,000,000	0.9799

## Data Availability

The data used to support the findings of this study can be made available by the corresponding author upon request.

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
