# Peer review of "Integration of Chemometrics and Sensory Metabolomics to Validate Quality Factors of Aged Baijiu (Nianfen Baijiu) with Emphasis on Long-Chain Fatty Acid Ethyl Esters"

_foods, 2023, doi:10.3390/foods12163087_

Round 1

Reviewer 1 Report

In this submitted paper, the authors tried to investigate the distribution of long-chain fatty acid ethyl esters (LCFAEEs) in Baijiu from different years (30 years) by combining modern flavor sensory analysis with multivariate chemometrics to comprehensively and objectively evaluate the impact of LCFAEEs on the quality of Baijiu. Some minor issues underlined in the following comments need to be resolved before further consideration.

Minor issues:

1.      P.4, L.116, the manufacturing city and country regarding the apparatuses ''Agilent Technologies 7890B GC System'' and the ''Agilent Technologies 5977A MSD'' mentioned here have to be provided.

2.      P.4, L.127, detailed information with regard to the apparatus used in ''mass spectrometry (MS) analysis'' including its commercial name, the manufacturing city and country must be indicated.

3. P.4, L.139, what do the authors mean by ''NIST''? Does it stand for a special full term?

Author Response

I sincerely appreciate your review. Your review has enhanced the quality of my article. Detailed modifications can be found in the attached document. Warmest regards.

Reviewer 2 Report

The manuscript entitled: “A 30-Year Tracking Study-Integration of Chemometrics and 2 Sensory Metabolomics to Validate Key Quality Factors of aged 3 Baijiu (Nianfen Baijiu)-Long Chain Fatty Acid Ethyl Esters” is scientifically sound and interesting. Please find my comments below that might help to improve the quality of this manuscript.

-I think, the authors should move some data in this article to the supplement files, for examples as Table 1, Figure 8 and so on.

-I would like to suggest the authors change the way to represent your data in Figure 1 into Table. As you can compare each compound of LCFAEEs in each year of Baijiu sample.

-The authors should add the Table that represents the relation between the distribution of 6 LCFAEEs in Baijiu of different years and four dimensions (i.e., visual sense, olfactory sense, taste sense, and evaluation after drinking)

-Table 3, I would suggest the authors should scoring the level indicator because as you presented with rating score 0-10 or 0-5, they were difficult to understand how to scoring because the wide range of rating!!

-I think the authors should conclude which compound of LCFAEEs that important for the quality of Baijiu. The new or old Baijiu, which one is the best? How to improve the quality of Baijiu?

-I think your work need to have ethics for research in human because you had the volunteers in your study to test the Baijiu, please add it in your article.

-Finally, I would say that your work is interesting, juts only your writing must emphasize the important data in the article and the others move to supplement data, it will be more interesting for reading.

-References must be following the journal format and must typing consistency.

Cheers!!!

Minor editing of English language required.

Author Response

(The authors gave the same response as above.)

Reviewer 3 Report

The present manuscript demonstrates the investigation of LCFAEEs in Beijiu beverage from biochemical and sensory perspectives. The changes of LCFAEEs in Beijiu during maturation were traced back to almost 30-year-old samples. The impact of LCFFAs on the sensory profiles of samples were tested using spiking and blending techniques. Data were analyzed and presented by means of chemometrics. The manuscript was very well prepared and written with a clear English language level. In my humble opinion, however, several minor suggestions as indicated below should be clarified.

·       Line 29-33: I recommend removing the suggestion for future research from the abstract to be mentioned in the final part of discussion.

·       Line 66: The term “evaluation after tasting” = aftertaste or not?

·       Line 98; Table 1: Why these six compounds were selected as key indicative LCFAEEs?

·       Line 116 / 127: The use of passive voice sentences is recommended.

·       Line 155-157: Reference(s) for LOD and LOQ determination should be addressed.

·       Line 159-167: I would recommend that this information is redundant and can be removed.

·       Line 176-200: I would recommend revising this part to be a more concise version. Redundant information should be avoided.

·       Line 286: the highest concentration of ET?

·       Line 263-298: I would recommend revising this part to be a more concise version. For example, the range of each compound detected in samples should be mentioned together with trends in their statistically significant levels among samples.

·       Line 304: Which compound was omitted from the analysis?

·       Line 305; Figure1: The caption panel of sample name can be removed since the names of samples have been already mentioned on the X-axis of all panels.

·       Line 310-312: Very interesting results. Why were all compounds detected in a lower concentration in B-1993 samples? What would be the possible explanation from the authors’ perspective?

·       Line 321-323: The concentration of only five compounds were subjected to the OPLSDA analysis and all of them are presented in the VIP scores. Normally, there should be a criterion for indicating that which compounds should be considered as potential biomarker responsible for discrimination based on their VIP scores, perhaps VIP > 1.0 and P < 0.05 or not, please find a reference regarding this issue. Also see line 467.

·       Line 328: In my humble opinion, I am not sure this is a common way to present Heatmap and cluster analysis for metabolomics data. What is the algorithm, i.e. type of correlation and linkage, used to construct the dendrogram?

·       Line 361-387: The biochemical insights regarding the development or dynamic changes of these aroma-flavor characteristics of Baijiu during aging should also be discussed.

·       Line 440-452: Very interesting concept for sensory work.

·       Line 470: Very interesting result.

·       Line 509: Potential implication of the findings in this study should also be mentioned.

·       Line 542-545: This part should be moved to the end of discussion.

Author Response

(The authors gave the same response as above.)

Reviewer 4 Report

General information.

The topic of the review article fits within the scope of the journal, and the article is well-written. The influence of not so much researched compounds in aged Baijiu and their potential effect (as added compound) is investigated to a great extent. Please address the following:

In Introduction section an explanation of Baijiu is needed. From what is it made and short description f production procedure.

Also, the general conclusion about the perfect age of Baijiu for the most effective concentration of LCFAEEs could be mentioned, I presume after 10-year period of aging (storage) (2013).

Modify the reference format according to the journal's requirements

The quality of the English Language is good. There are some spelling errors that can be corrected easily.

Author Response

(The authors gave the same response as above.)

Reviewer 5 Report

Manuscript ID Foods-2540326

Titled: " A 30-Year Tracking Study——Integration of Chemometrics and 2 Sensory Metabolomics to Validate Key Quality Factors of aged 3 Baijiu (Nianfen Baijiu)——Long Chain Fatty Acid Ethyl Esters."

Comments to the Author:

The manuscript addressed to the important topic about the influence of aging process on the sensory quality of Baijiu according to the objectives described in lines 88-93. Currently, the consumers highly value quality, originality and diversity of beverages. Baijiu is a very popular drink in China. However, despite being the best-selling distillate in the world, it is little known outside China. Nevertheless, the authors present a comprehensive bibliography on this liquor, so much so, the work may be interesting for aims and readers of Foods but should be improved in some critical aspects, especially the related to the results.

TITLE, ABSTRACT AND KEYWORDS

The title does not agree with the content of the article. It is not a follow-up study over 30 years, as the authors analyze a sample from thirty years ago and seven more samples produced in 2003, 2008, 2013, 2014, 2015, 2017 and 2018.

The abstract is in accordance to the content of the article.

INTRODUCTION

The introduction is well organized. The authors describe the problem and what they hope to achieve. The overall presentation is clear and concise.

MATERIALS AND METHODS

Sampling

The description of samples and sampling is very laconic. As stated in line 101, the samples were obtained from a single distillery, and as shown in Table 2 eight groups of samples were obtained. There is no reference to how many samples from each group were analyzed. It appears as if it was a single sample per group. If this is so, in my opinion it seems an insufficient number to meet the proposed objectives and draw valid conclusions.

The methodology used for the implementation of the study is consistent with the aim of this work and is correctly applied. The description of methods is exhaustive and the methods are well explained, except the Statistical analysis. Many statistical methods are named but little information is provided about them. PCA method is named but has not been used.

Line 116. I think that the information about conditions of splitlees injection should be included.

Section 2.6.

The authors describe how they obtain all analytical quality parameters except recovery. In my opinion this information should be included, explaining whether they use standards or spiked samples and how they do it.

Table 2.

For more clarity I would put dividing lines so that each descriptor is clearly visible.

Some points are missing at the end of the descriptions.

RESULTS AND DISCUSSION

In general, this section is mainly based on results. Very little discussion with other authors is brought up, especially in the case of LCFAAEs.

In my opinion it would have been interesting to compare the obtained results for various types of samples obtained from similar raw materials.

line 246. I believe that the precision values should be less than 3% and 9% instead of 5% and 10%.

Table 4.

The interday and intraday precision columns are interchanged according to the paragraph of line 246.

If possible, the authors should explain why the recovery rate results are expressed as an interval. Is it the confidence interval? If so, they should indicate.

Section 3.2

This section is very confused. Figure 1 should be named first to focus the reader's attention. The results are not well explained and they are difficult to understand. For example, I suppose that when the authors talk about statistically significant differences, they have tested it by ANOVA or another statistical test. This question should be clarified, and the authors should refer to the technique used.

Line 284 and 285. The sentence " The concentration of ET varied between 0.33 and 0.52 mg/L across different sample groups." should be placed in the following paragraph.

Line 292. The authors say : “EO only reached the quantitation limit in 4 Baijiu sample groups with longer storage time, indicating a notable point”, but the LOQ values (Table 4) for EO is 1.13 higher than the values given for the samples. This should be clarified.

Line 321.  “ and 912-EO” instead of “912-EO and”

Figure 2. Figures b and c are interchanged.

Section 3.3

If possible, the authors should explain why the results of the sensory analysis are presented with different types of graphs. They should also clarify whether the results obtained have been contrasted by ANOVA or similar technique. If so, it should be reflected in the corresponding figures.

The authors state that there are significant differences or similar on several occasions, but there is no reference to the test used (lines 346, 362, 398, 404, 407, 418, 423)

Lines 367 and 368. The authors should explain more extensively how they have done the correlation analysis: Pearson, Spearman, …?

They also say: " whereas no significant correlation was found with other aroma dimensions (Figure 4)” but Figure 4 shows no correlations between LCFAEEs and aroma descriptors".

Lines 382-386. I don't understand this paragraph. Does this study confirm that LCFAEEs do not contribute to Baijiu aroma?

Figura 4.

To better visualize the information in figures a-e, they should put only one legend since it is the same for all figures.

Lines 418-421.  This paragraph should be clarified. I do not understand the claims made by the authors: “By examining the concentration of LCFAEEs in different sample groups, obviously, the concentration of LCFAEEs was positively correlated with the perception of irritation (r = 0.429)”. I don't see it as obvious. It is also not clear to me that it can be deduced that LCFAEEs have a positive effect on the softness of Baijiu.

Line 422.  B-1993 has lower scores for off-flavor compared to other sample groups, except B-2015 , as can apparently be seen in Figure 6.

Section 3.4

The authors limit themselves to draw conclusions and present graphs but do not explain the reasons for their statements, and some of them are not so evident, at least from my point of view (lines 458-468).

Lines 483-485.  The authors should give more information about figure 8a in order to understand how the incorporation of LCFAEEs influences taste and evaluation after drinking of the Baijiu

Lines 491-493. The authors should clarify this statement: “By comparing the data from the individual addition of LCFAEEs (Figure 8g, h, i, j, k, l), it was found that EP and 912-EO had a more remarkable effect on the afore mentioned characteristics.” Figure 8g shows that compared to the original beverage, the presence of ET decreases the off-flavors by 25% more than any other compound.

REFERENCES

Reference 6 and 56 are the same.

Author Response

(The authors gave the same response as above.)
